# Unexpected dynamics in femtomolar complexes of binding proteins with peptides

Stefano Cucuzza[1], Malgorzata Sitnik [1], Simon Jurt[1], Erich Michel[1,2], Wenzhao Dai[1], Thomas Müntener [3], Patrick Ernst [2], Daniel Häussinger [3], Andreas Plückthun [2] ✉ & Oliver Zerbe [1] ✉

Ultra-tight binding is usually observed for proteins associating with rigidified molecules. Previously, we demonstrated that femtomolar binders derived from the Armadillo repeat proteins (ArmRPs) can be designed to interact very tightly with fully flexible peptides. Here we show for ArmRPs with four and seven sequence-identical internal repeats that the peptide-ArmRP complexes display conformational dynamics. These dynamics stem from transient breakages of individual protein-residue contacts that are unrelated to overall unbinding. The labile contacts involve electrostatic interactions. We speculate that these dynamics allow attaining very high binding affinities, since they reduce entropic losses. Importantly, only NMR techniques can pick up these local events by directly detecting conformational exchange processes without complications from changes in solvent entropy. Furthermore, we demonstrate that the interaction surface of the repeat protein regularizes upon peptide binding to become more compatible with the peptide geometry. These results provide novel design principles for ultra-tight binders.

Binding of rigid ligands to proteins is believed to proceed via an initial encounter into a more or less preformed binding site[1]. There is usually a unique way in which the ligand forms contacts to the protein. The incoming ligand may display shape complementarity to the binding pocket, and binding then proceeds via small relative movements as described by the key-and-lock model. Alternatively, structural adaptations of the receptor may be required, either by selecting one out of many existing receptor conformations (conformational selection) or via ligand-induced structural adaptions after an initial encounter (induced fit)[1–4]. In case of flexible ligands structural adaptations may occur both in the ligand as well as in the receptor. Despite the dynamics associated with these adaptations the ligand is generally assumed to be rigid once held in the binding pocket. Binding affinity in a series of potential ligands is increased by providing a sufficiently large number of polar and/or apolar contacts and optimizing both surface complementarity and the orientation of polar groups to form intermolecular hydrogen bonds[4]. All these modifications optimize binding enthalpy, but essentially neglect the entropic contribution to binding. As a result, the best binders most often only reach the low nanomolar regime because entropic losses, by freezing multiple degrees of freedom upon binding larger molecules, mitigate the gain of additional enthalpic interactions[5].

Here we challenge the view that rigidifying molecules to reduce entropic losses upon binding is always the best way for obtaining very tight binders. To this end we introduce a series of peptides that are fully flexible in their unliganded states, and that can associate with the receptor in various ways. To accomplish this, we chose the ligand and the target protein to have repetitive epitopes and binding pockets. The manifold of potential binding modes, including some where individual side chains have left their pocket, may then introduce entropic effects that could influence binding affinity and kinetics. To investigate this phenomenon in more detail, we present here a systematic study of structural and kinetic aspects of flexible ligand recognition in a system that comprises highly symmetric binding sites, and that is recognized

[1]Department of Chemistry, University of Zürich, Winterthurerstrasse, 190, 8057 Zürich, Switzerland. [2]Department of Biochemistry, University of Zürich, Winterthurerstrasse, 190, 8057 Zürich, Switzerland. [3]Department of Chemistry, University of Basel, St. Johanns-Ring 19, 4056 Basel, Switzerland. ✉e-mail: plueckthun@bioc.uzh.ch; oliver.zerbe@chem.uzh.ch

by ligands that share the corresponding symmetry in their binding epitopes. Importantly, some of these peptide-protein complexes comprise extended binding interfaces that allow for partial unbinding without ligand dissociation and thus introduce ligand dynamics in the bound state.

In our study we utilized recently developed designed consensus-derived Armadillo repeat proteins (dArmRPs) that contain internal modules, termed "M", capable of recognizing Lys-Arg (KR) dipeptides[6–8]. The dArmRP consist of stacked sequence-identical modules (M), structurally characterized by 3 α-helices arranged in a triangular arrangement. The internal M modules are flanked by N-terminal ("N") and C-terminal ("C") capping repeats that shield internal modules from the aqueous phase, and they usually do not participate in ligand binding. In an idealized way, a $NM_4C$ dArmRP can bind a $(Lys-Arg)_4$ octapeptide via its four internal modules[9]. Of note, in this system the peptide may shift, such that different parts of the peptide may bind to different parts of the protein, resulting in a multitude of different potential binding modes. Furthermore, since the overall binding affinity is very high (see below) individual side chains might temporarily leave their pocket without the peptide dissociating. Using fluorescence polarization assays we demonstrated that each additional repeat contributes similarly and in an additive fashion to the overall binding energy[9]. Consequently, the equilibrium association constants can be approximated by multiplication of the corresponding constants resulting in pico- or even femto-molar equilibrium dissociation constants for dArmRPs with 7 or more internal repeats[8].

Herein, we investigate, using solution NMR techniques, binding of a series of peptides of the general type $(Lys-Arg)_n$ to dArmRP containing x internal modules for cases in which the number of dipeptide units n is smaller, equal or larger than the number of binding modules x. We use chemical shift perturbation data (CSPs) to follow the dynamics of peptide binding, and determine the structural response of the protein to peptide binding from pseudocontact shifts (PCS). Given the low equilibrium dissociation constants for some of these complexes, indicating very tight overall binding, transient loss of some of the interactions may not necessarily result in complete disintegration of the protein-ligand complex. We observed that, despite their very high binding affinities, these systems are dynamic. Using NMR spectroscopy we develop a very detailed picture of the binding dynamics that is complementary to other biophysical techniques such as surface plasmon resonance (SPR)[10] or fluorescence polarization assays[11] that monitor only complete binding or unbinding processes. Moreover, we discovered that binding in solution is more dynamic and less regular compared to the states captured in crystal structures, in which packing forces likely influence and regularize the binding mode. The data will contribute to the understanding of peptide binding to proteins and thereby facilitate successful protein design.

## Results

The aim of this study is to investigate binding of peptides to modular proteins. We studied peptide recognition and binding mode, conformational dynamics of the bound state and structural adaptations of the protein upon peptide binding. To this end we apply solution NMR techniques that allow to characterize the system with atomic resolution and in solution, devoid of potentially interfering crystal contacts that are present in the solid state. We use chemical shift perturbations (CSPs) to locate the binding sites and follow kinetics of peptide binding under equilibrium conditions. We then investigate how the protein $NM_4C$ structurally adapts to the binding of the peptide $(KR)_4$ by computing structures of free and peptide-bound proteins from pseudocontact shifts (PCSs)[12]. Finally, we describe binding of $(KR)_n$ to $NM_xC$ with $n = 4, 5, 6$ and $7$ and $x = 4$ and $7$. We looked at two versions of the proteins with different N-caps, the so-called yeast-derived $N^Y$ cap[6,13] and the recently developed more stable $N^A$ cap[14]. To accomplish the challenging task of chemical shift assignments in proteins composed of a repetitive amino acid sequence we applied segmental isotope labeling using expressed protein ligation.

## Chemical shift assignments

The highly repetitive nature of dArmRPs sequences render sequence-specific resonance assignments by classic triple-resonance experiments impossible. To overcome this issue, we resorted to a combination of biochemical and spectroscopic methods. We combined information from segmental isotope labeling (Supplementary Fig. 1) through expressed protein ligation (EPL)[15, 16], chemical shift assignments previously derived by us for a $N^YM_2$:MC protein complex[17], and verified assignments from the series of proteins with an increasing number of internal repeats $NM_xC$ with $x = 1–4$ (Supplementary Fig. 2). The assignment strategy used for $N^YM_4C$ is summarized in Fig. 1B. Since binding of the ligand improved signal dispersion, and in particular lifted chemical shift degeneracies in the different internal modules, the $(KR)_4$-bound state was assigned first. Assignments were subsequently adapted to the apo state by tracing signals back from two to zero equivalents (Fig. 1C).

In EPL the N-terminal partner is fused to an intein which, when mixed with the C-terminal fragment, is ligated to the latter in a process known as protein splicing[15, 18]. The full $N^YM_4C$ construct was separated into two segmentally labeled constructs, $N^YMab$ and $cM_2C$ as well as $N^YMMab$ and $cMC$, where a, b and c represent helix 1, 2 and 3 of an internal module, respectively (Supplementary Fig. 1), both resulting in an $N^YM_4C$ molecule. Each of these two constructs was made twice, once with labeling of the N-terminal and once with labeling of the C-terminal fragment. Tentative side chains assignments used these segmentally labeled proteins and were made based on those from the $N^YM_2$:MC complex, which were then validated using HCCH-TOCSY and $^{13}C$-resolved NOESY experiments. Finally, assignments were corroborated by comparing chemical shifts with those in the series $N^YM_xC$ where $x = 1, 2, 3$ or $4$. $N^YM_7C$ was assigned via a similar approach. Either $N^YMMab$ or $cMC$ segments of the full-length protein were labeled by EPL (Supplementary Fig. 1). Despite the increased spectral complexity resulting from the addition of three more identical internal repeats, $N^YMMab$ and $cMC$ were almost fully assigned. For more details on assignments see the SI.

## Binding dynamics of $(KR)_n$-type peptides to $NM_4C$ and $NM_7C$

Next, we studied binding dynamics of peptides of the type $(KR)_n$ to dArmRP proteins with 4 or 7 sequence-identical internal modules by monitoring chemical shift perturbations (CSPs) via $[^{15}N,^1H]$-HSQC experiments. The peptides contained 4–7 KR dipeptide units. When comparing the length of the peptide to the length of the protein's binding surface, they can be grouped as matching (i.e. $NM_7C + (KR)_7$), longer (i.e. $NM_4C + (KR)_6$) or shorter (i.e. $NM_7C + (KR)_4$). In the case of $(KR)_4$ binding to $NM_4C$ we additionally studied the structural adaptations in the protein by refining the structures of free and peptide-bound $NM_4C$ by pseudocontact shift (PCS)-derived restraints using methods described by us recently[19].

Peptide binding to ArmRP results in many changes in the spectra. In our description below we particularly focus on changes for reporter residues, i.e. residues that are directly involved in binding, as known from crystal structures, or those that report on structural changes. Residues directly involved in binding are located in H3 of each module, in particular the Trp residues that form cation-π interactions with the guanidino group of Arg residues. Previous crystallographic studies suggested that differences in the protein upon binding mostly occur in the supercoil[20]. Changes in the supercoil affect residues in the loop connecting neighboring repeats while leaving the modules themselves virtually unchanged. Gly residues from these loops are located in distinct, less crowded regions of the spectra, and were therefore used to monitor the structural changes. Not surprisingly, signals resulting from glycines of the type

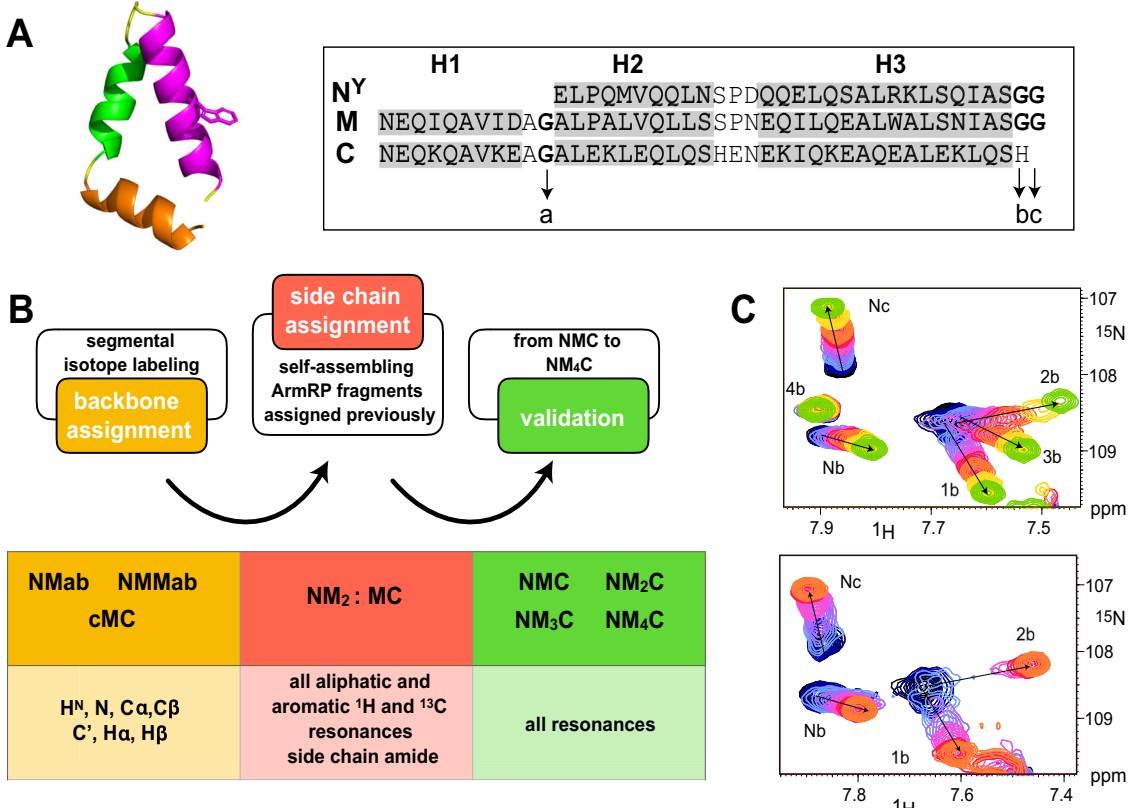

**Fig. 1 | Assignment procedure for dArmRPs. A** Structure of a single ArmRP module with H1 in orange, H2 in green and H3 in purple, and sequences of $N^Y$, M and C modules. Note that $N^Y$ does not have helix H1. Glycine residues are listed as) (**a–c**). **B** Schematic representation of the $N^YM_4C$ assignment strategy. **C** Expansions of [$^{15}$N,$^1$H]-HSQC spectra of uniformly (top) and $N^Y$MMab-labeled (bottom) $N^YM_4C$,

showing selected cross-peaks from glycine residues from the $N^Y$-caps and internal modules. For definition of the fragments for segmental labeling see Supplementary Fig. 1. Contours of spectra in the presence of 0-2 equiv. of peptide are color-coded differently (from dark blue (0 equiv.) over orange (1 equiv.) to green (2 equiv.)). The direction of peak shifts is indicated by arrows.

*a* (cf. Fig. 1B), located in the H2-H3 loop, either do not move or shift only very little upon binding of the peptide and hence were excluded from further analysis. In contrast, glycines of the type *b* and *c*, located in the H3-H1 loop that connect an internal repeat to the next one, are sensitive to changes in the relative orientation of neighboring modules (Fig. 1B; Supplementary Fig. 3). They were observed to directly participate in the binding in only one crystal structure (there, type *b* formed hydrogen bonds between their O atoms and ε-amino groups of the Lys side chain of the peptide)[9].

Biochemical data indicated that the original **y**east-derived $N^Y$-terminal cap possesses an intrinsic instability against *E. coli* proteases, likely due to insufficient packing of the cap against the M1 module. The redesigned N-cap, described in Michel et al.[14] and referred to as $N^A$ (**a**rtificial), displayed improved stability and yielded very similar CSPs. A comparison of CSPs between $N^YM_4C$ and $N^AM_4C$ upon binding (KR)$_4$ revealed similar binding for both constructs (Supplementary Fig. 4), and affinity determination by FP resulted in a highly similar $K_d$ to the peptide (KR)$_5$: 36.1 ± 2.9 nM for $N^YM_4C$ versus 30.5 ± 2.3 nM for $N^AM_4C$ (mean ± S.D.)[14]. This combined evidence clearly demonstrates that the two protein variants respond almost identically to peptide binding, and data below therefore mostly describe structures employing the new N-cap. Assignments for $N^AM_4C$ were adapted from $N^YM_4C$ using simple triple-resonance experiments. Unfortunately, it was impossible to adapt assignments of $N^AM_7C$ without further segmental labeling.

### Binding dynamics of (KR)$_n$-type peptides to $N^{Y/A}M_4C$
We first describe binding dynamics for the case when the number of dipeptide units matches the number of modules. In the apo state the

chemical environment of signals from internal repeats are very similar. For example, in $NM_4C$ the four NH signals of the Trp indole moieties are almost perfectly superimposed (Fig. 2A). Titrating (KR)$_4$ to $N^AM_4C$ resulted in spectra displaying features of fast and intermediate exchange on the NMR timescale, in agreement with the observed $K_d$ of 265 nM[9]. For example, the indole signals shift upon addition of peptide, corresponding to the fast exchange regime, but are very broad between 0.4 and 1 equiv. of peptide. Above 1 equiv. the peaks from M1 and M4 appear while the signals from M2 and M3 are only visible at full saturation. Interestingly, indole signals from the two internal modules that pack against the caps, M1 and M4, experience a different environment, resulting in signals separate from those of M2 and M3 also in the $^1$H dimension (Fig. 2A). Similarly, 7 out of the 10 b- or c-type Gly amide signals move steadily from zero up to 1.3 equivalents of the peptide, after which the system is fully saturated (Fig. 2B, Supplementary Fig. 3). Similar as the indole signals, the three Gly signals 1c, 2c and 3c display very broad peaks in the first steps of the titration indicating intermediate exchange.

The largest CSPs upon peptide binding are located mostly in helix 3 around the canonical binding pocket[9] of internal modules (Fig. 2C and Supplementary Fig. 4), with small or no CSPs in the N- and C-terminal caps. Surprisingly, moderate-to-large CSPs were also detected in the loop connecting $H_3$ of a given module to $H_1$ of the following one and in the center of $H_2$. As these regions are not directly involved in peptide binding, they rather report on conformational adaptations of the protein upon peptide binding. Since crystal structures of dArmRP in peptide-bound-states display no significant changes within the modules[20], we suspected

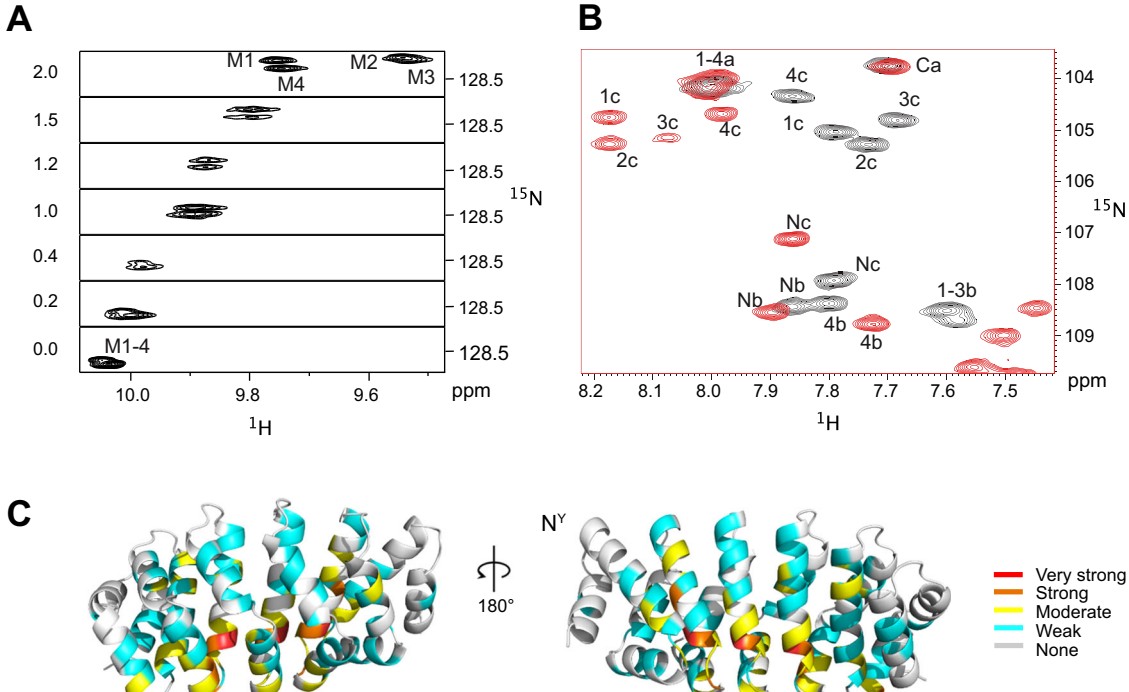

**Fig. 2 | Binding of (KR)$_4$ to N$^Y$M$_4$C.** Expansions of [$^{15}$N,$^1$H]-HSQC spectra of indole Hε (**A**) and Gly amide (**B**) region of N$^Y$M$_4$C upon addition of (KR)$_4$. Assignments in the apo and bound states are indicated by the number of the internal module. In (**B**) black contours refer to the apo-protein, while red contours refer to the (KR)$_4$-saturated state with 1.3 equivalents of peptide. **C** Chemical shift perturbations of

N$^Y$M$_4$C upon binding of (KR)$_4$ color-coded onto the structure using the following color code: very strong−red (>1 ppm), strong−orange (<1.0 and >0.5 ppm), moderate−yellow (<0.5 and >0.25 ppm), weak -cyan (<0.25 and >0.11 ppm) and none−gray ( < 0.11 ppm).

that the observed changes are rather due to alterations of the supercoil.

## Structural adaptations of N$^A$M$_4$C upon binding (KR)$_4$

Next, we investigated how the protein responds to peptide binding. In principle, this can be determined by crystallography. However, repeat proteins form crystal contacts that might impact the supercoil[9, 20–23], and hence we resorted to solution NMR techniques. To bypass problems related to sidechain assignments in repeat proteins, required for NOE-based methods, we developed an automated iterative approach for protein structure refinement using pseudocontact shifts (PCS)[19]. In brief, structural information was extracted from a crystal structure and used to restrain each module to loosely conserve the overall triangular structure of individual internal modules and caps. Four slightly different starting models, generated from the crystal structure of N$^Y$M$_5$C (PDB ID: 5MFN)[20] as described in the SI and in Cucuzza et al.[19], were iteratively refined based on experimental PCS to define the supercoil of the protein. To this end N$^A$M$_4$C was tagged with lanthanide-binding tags (LBTs)[24–27] at uniquely introduced Cys residues in three positions, triggering PCS, and chosen to be equally spread throughout the molecule without interfering with peptide binding: (C)S21C, (M2)Q18C and (N$^A$)E15C, where numbering in the parenthesis refers to the capping or internal repeat that carries the Cys residue. PCSs were then extracted either in the apo or (KR)$_4$-bound states (Supplementary Fig. 5, Supplementary Data 1).

The resulting structures in both states after ten cycles of refinement are shown in Fig. 3A. Changes of the supercoil upon binding of (KR)$_4$ were analyzed employing the Rosetta symmetry framework (see Materials and Methods) by monitoring the Cα(P/P + 2) distance[8], a parameter that represents the configuration of a peptide bound to two neighboring internal modules (Fig. 3B).

The Cα(P/P + 2) distance suitable for modular binding of extended peptides is in the range of 6.7-7.0 Å[8]. The apo state displays Cα(P/P + 2) values (mean ± S.D.) at the termini of the binding surface as large as 9.00 ± 0.19 Å for M1-M2 and 9.82 ± 0.38 Å for M3-M4, while a smaller value of 6.82 ± 0.19 Å is observed for the central pair M2-M3 (Fig. 3B).

Binding of (KR)$_4$ triggers changes in protein curvature that result in a gentle regularization of the binding surface by reducing values (mean ± S.D.) at the termini to 8.14 ± 0.35 Å (M1-M2) and 8.69 ± 0.24 Å (M3-M4), while increasing those for the central pair M2-M3 to 7.86 ± 0.25 Å (Fig. 3C). While a decrease in Cα(P/P + 2) at the edges of the binding surface is consistent with the expected changes for extended modular binding[20], the central pair surprisingly displayed the opposite behavior. However, if protein-peptide contacts are continuously broken and reformed, and hence the requirement for forming all contacts simultaneously is relaxed, regularization of the binding surface as observed would be expected (vide infra), and in this system, they would not necessarily converge to the value optimal for continuous binding.

## Binding of longer peptides to larger proteins

Next, we investigated the binding of (KR)$_7$ to NM$_7$C (Fig. 4A, B). For most of the peaks we observed a buildup of a relatively sharp second set of peaks (corresponding to M1, M7 and M2-M6) with an intensity that follows the proportion of added peptide, as expected considering the sub-picomolar $K_d$ that should clearly place exchange into the slow regime. Full saturation is reached after adding 1.0 equiv. of peptide. CSPs for Gly signals follow similar trends as for NM$_4$C where signals from Gly in the back of the binding surface (Gly-a) move very little, while those in the loops between modules (Gly-b/c) are more strongly shifted, in particular those from the central modules 2-6 (Supplementary Fig. 11). A similar behavior was observed for the indole signals (Supplementary Fig. 14).

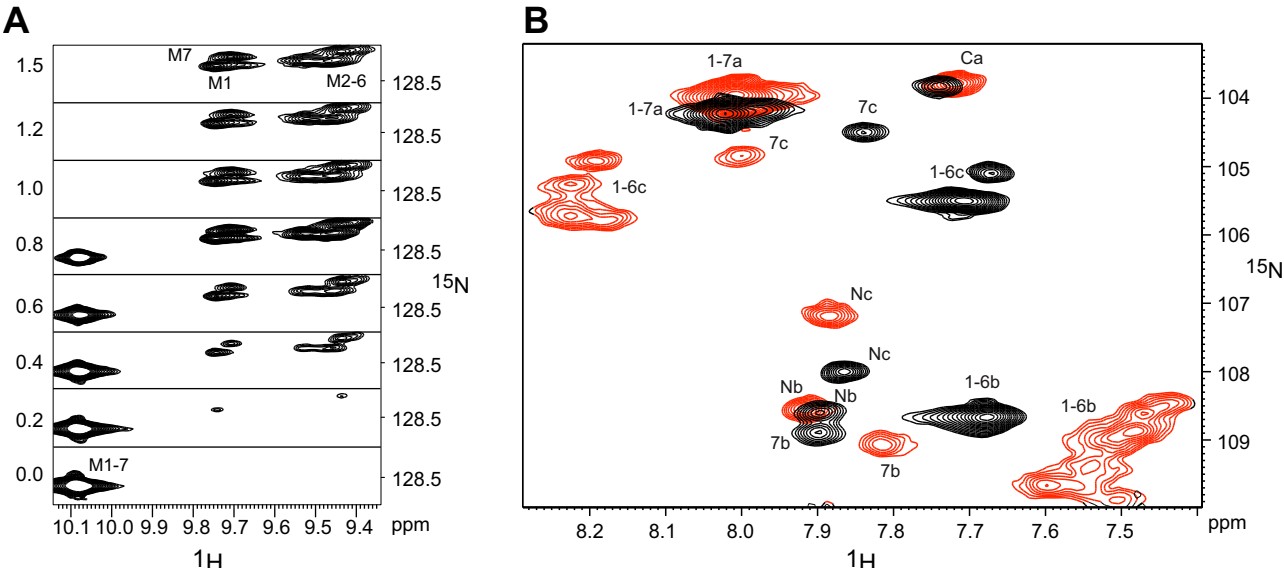

**Fig. 3 | Structural adaptations of $N^AM_4C$ upon binding $(KR)_4$. A** $N^AM_4C$ structures, obtained in this study by CYANA's PCS refinement starting from a representative model, in the apo (green) or $(KR)_4$-bound (red) states aligned across the entire sequence. The structure at the bottom is obtained by a 180 degrees rotation about a vertical axis. **B** Characteristic distances of atoms from neighboring repeats in the apo (green bars, black dot) and peptide-bound (red bars, orange diamond) states, represented as mean ± S.D. ($n = 4$). The definition of the $C\alpha(P/P+2)$ distances, depicted on top, is related to the supercoil and measured as the distance of two $C\alpha$ atoms of residues i, i + 2 of a stretched peptide bound in the canonical mode[8].

**Fig. 4 | Binding of $(KR)_7$ to $N^YM_7C$.** Expansions of [$^{15}N$,$^1H$]-HSQC spectra of indole Hε (**A**) and Gly amide (**B**) region of $N^YM_7C$ upon addition of $(KR)_7$. **A** Expansions of the indole region of spectra at different molar equivalents of peptide (indicated by the number on the left). In (**B**) black contours refer to the apo state, while red contours refer to the $(KR)_7$-saturated state. Assignments in the apo (black) and bound states (red) are indicated by the number of the internal module.

## Dynamics of peptide binding

Previously, we discovered that free energies of binding (ΔG) for (KR)-type peptides to ArmRPs depend in an additive fashion on the number of internal modules[9]. Dissociation constants for binders with $K_d s < 1\,nM$ were obtained by interpolation of the data from shorter dArmRPs. In order to directly assess the binding constants for dArmRPs with seven internal repeats, a FRET-based assay was used that in addition allowed to derive the kinetic constants $k_{on}$ and $k_{off}$[28]. Tab. 1 clearly demonstrates the stepwise increase in affinity to NM$_7$C for each additional (KR) unit in the peptide.

Finally, we aimed at determining the binding dynamics at atomic resolution by comparing the measured line-shapes of the peaks with simulations (Fig. 5). Considering the low $K_d$ of about 300 nM for (KR)$_4$ binding to NM$_4$C as determined by FP[9] one would expect very little exchange between bound and unbound species at saturation, and hence no additional line-broadening on the bound-state signal. Surprisingly, experimentally observed line-shapes are much broader than simulated ones when using a two-state model with on- and off-rates such that their ratio is consistent with the $K_d$ of 300 nM. For a reasonable fit, a second exchange process involving the peptide-bound

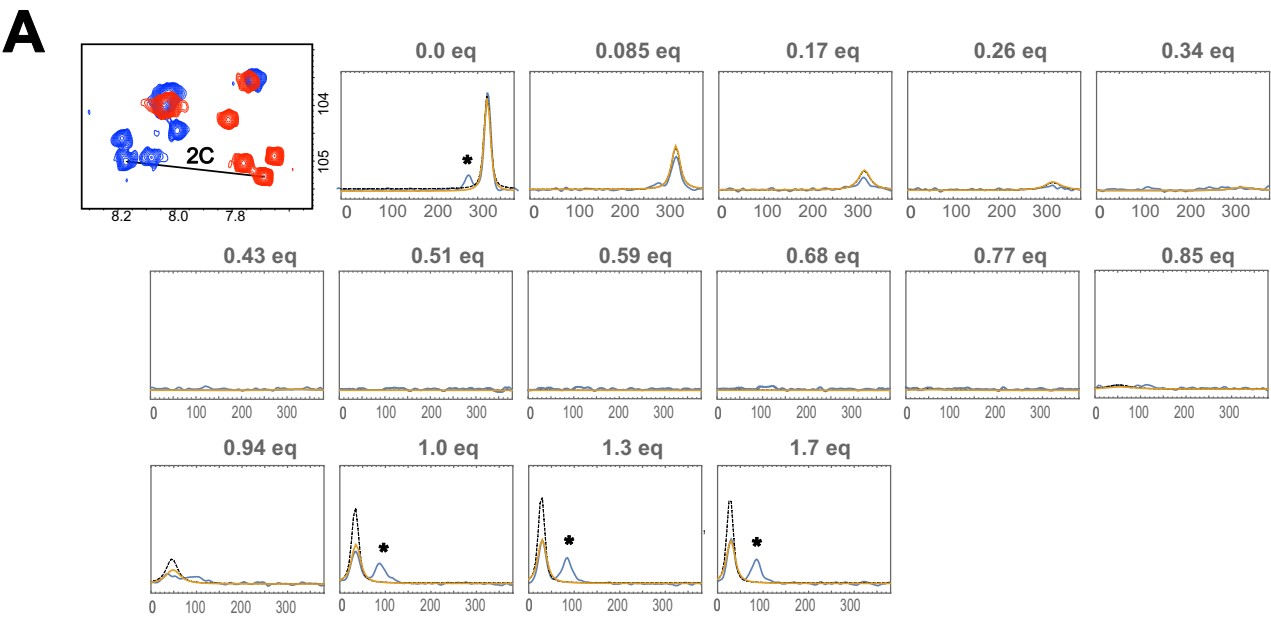

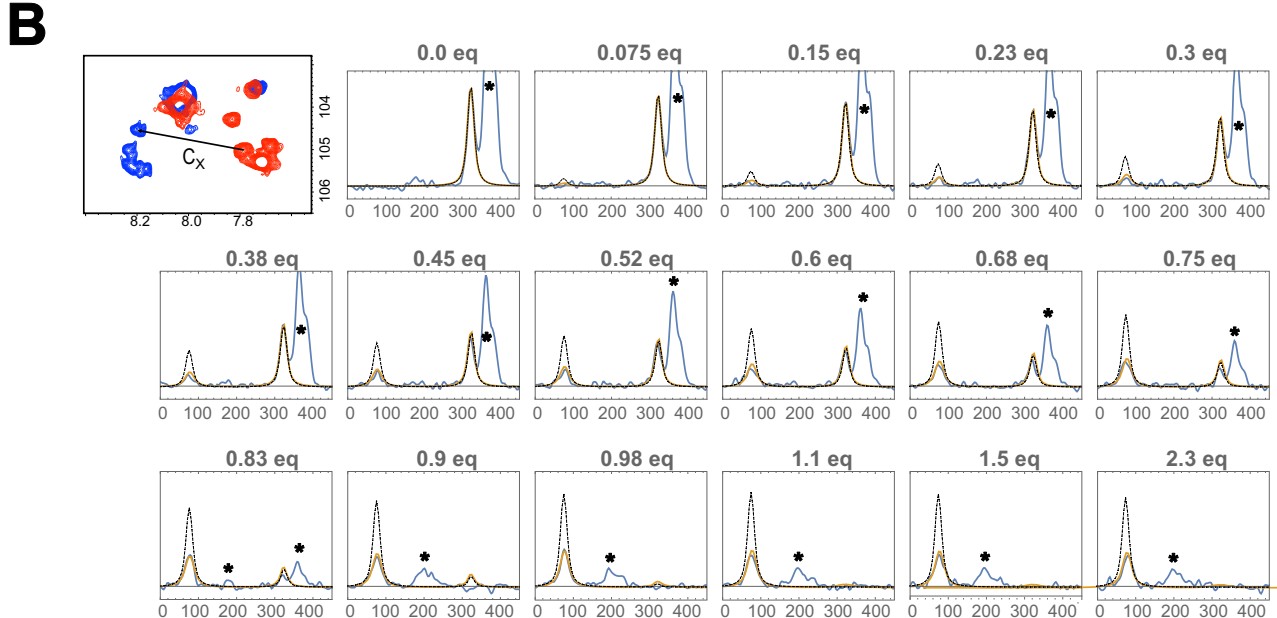

**Fig. 5 | Simulations of line-shapes and comparison with those obtained during the titration with peptides. A** N$^A$M$_4$C-(KR)$_4$ or (**B**) N$^A$M$_7$C-(KR)$_7$. Traces are taken from peaks of [$^{15}$N,$^1$H]-HSQC spectra during titrations with the corresponding peptides. Simulation parameters are (**A**): $k_{1,on} = 10^9\,M^{-1}\cdot s^{-1}$, $k_{1,off} = 300\,s^{-1}$, $k_2 = 30\,s^{-1}$, $k_{2'} = 700\,s^{-1}$, $\Delta\omega = 294\,Hz$. **B** $k_{1,on} = 6.2\cdot10^8\,M^{-1}\cdot s^{-1}$, $k_{1,off} = 0.0003\,s^{-1}$, $k_2 = 30\,s^{-1}$, $k_{2'} = 100\,s^{-1}$, $\Delta\omega = 246\,Hz$. Expansions of the corresponding spectra for initial (red contours) and final (blue contours) states are depicted on the left. Experimental and fitted lines with the additional isomerization step are depicted with blue and orange lines, respectively. Line-shapes obtained from a two-state fit, i.e. without isomerization in the bound state ($k_2 = 0$), are depicted with black dotted lines for comparison. The peaks denoted with an "*" are from different Gly residues.

species with a much different time scale for the rate constants was required, e.g. $k_2 = 30\,s^{-1}$ and $k_{2'} = 700\,s^{-1}$ (compare solid and dotted lines in Fig. 5A):

$$P + L \underset{k_{1,off}}{\overset{k_{1,on}}{\rightleftharpoons}} PL \underset{k_{2'}}{\overset{k_2}{\rightleftharpoons}} PL^* \tag{1}$$

Note that the line shapes cannot be fitted unambiguously as the chemical shift of the fully bound state is not known. Simulations, however, demonstrate that the position of the bound state signal must be very near to the observed position, and hence $k_2$ and $k_{2'}$ are likely close to the above given values (Supplementary Fig. 12). We believe that the first exchange process accounts for overall binding of the peptide to the dArmRP, i.e. the process measured by FP, while the second process corresponds to local exchange processes, in which individual contacts are transiently broken.

In case of $(KR)_7$ binding to $NM_7C$ signals for the bound state are even broader, although the dissociation constant is significantly smaller (Fig. 5B). To reproduce the dissociation constant of 484 fM, a $k_{1,on}$ of $6.2 \cdot 10^8\,M^{-1}\,s^{-1}$ and a $k_{1,off}$ of $0.0003\,s^{-1}$ was used. The second exchange process is best described by a $k_2$ of $30\,s^{-1}$ and a $k_{2'}$ of $100\,s^{-1}$.

### Binding dynamics of $(KR)_n$-type peptides to $NM_xC$: The cases of mismatch

$NM_4C$ titrations with peptides, where the number of dipeptide units and the number of internal repeats ($(KR)_5$, $(KR)_6$ or $(KR)_7$) do not match, resulted in spectra again displaying features of both fast and intermediate exchange (Supplementary Fig. 13). The main difference for the four peptides $(KR)_4$ to $(KR)_7$ binding to $N^AM_4C$ is a slightly decreased amount of peptide required for full saturation (1.3 equiv. $(KR)_4$ vs 1.0 equiv. $(KR)_7$). In addition, Trp indole protons remained very broad for $(KR)_6$ and $(KR)_7$ but much less for $(KR)_5$ (Supplementary

Fig. 13) indicating the presence of intermediate exchange in the saturated states. Binding affinities are higher for peptides with more dipeptide units than the number of internal modules in the target dArmRP, in comparison to peptides with fewer dipeptide units[9].

There are several factors which all may contribute to this observation: First, the increased number of binding states increases the likelihood of binding, and second, there might be additional long range electrostatic effects from the extra Arg or Lys residues. During titrations of $N^AM_7C$ with $(KR)_4$ and $(KR)_5$ bound-state signals display again features of intermediate exchange (Supplementary Fig. 14). The linewidths of the bound-state signals decrease when going from $(KR)_4$ to $(KR)_5$ to $(KR)_6$. Corresponding titrations of $N^YM_7C$, which possesses the unstable $N^Y$ cap, with $(KR)_4$ display a slightly different behavior: Here, slow exchange is observed only up to 1 equiv. of $(KR)_4$ for the Trp indole signals, after which signals follow further changes in the fast-exchange regime (Fig. 6A). $N^AM_7C$ contains seven $(KR)$ binding sites of which only four are occupied by $(KR)_4$. The extra three empty sites are therefore available for a second molecule, which is expected to bind with much lower affinity, placing it into the fast-exchange regime. Binding of $(KR)_4$ to $NM_7C$ hence is a two-step process with an initial tight binding of the first peptide followed by weaker binding of a second peptide.

Interestingly, the bound-state indole signals are much broader for the $N^AM_7C:(KR)_4$ complex than for the $N^YM_7C:(KR)_4$ complex (Supplementary Fig. 16). We suspect that peptide binding in the protein with the less stable N-cap occurs preferably towards the C-terminus, while in the protein with the more stable N-cap various binding modes are thermodynamically similar and hence rapid exchanges between various modes of binding occur (vide infra). We therefore investigated whether primary and secondary binding events occur at preferred sites. To this end we produced $N^YM_7C$ segmentally labeled at the $N^YMMab$ and $cMC$ termini by EPL (Supplementary Fig. 1), and determined CSP data upon addition of $(KR)_4$ (Fig. 6B). CSPs are largest for

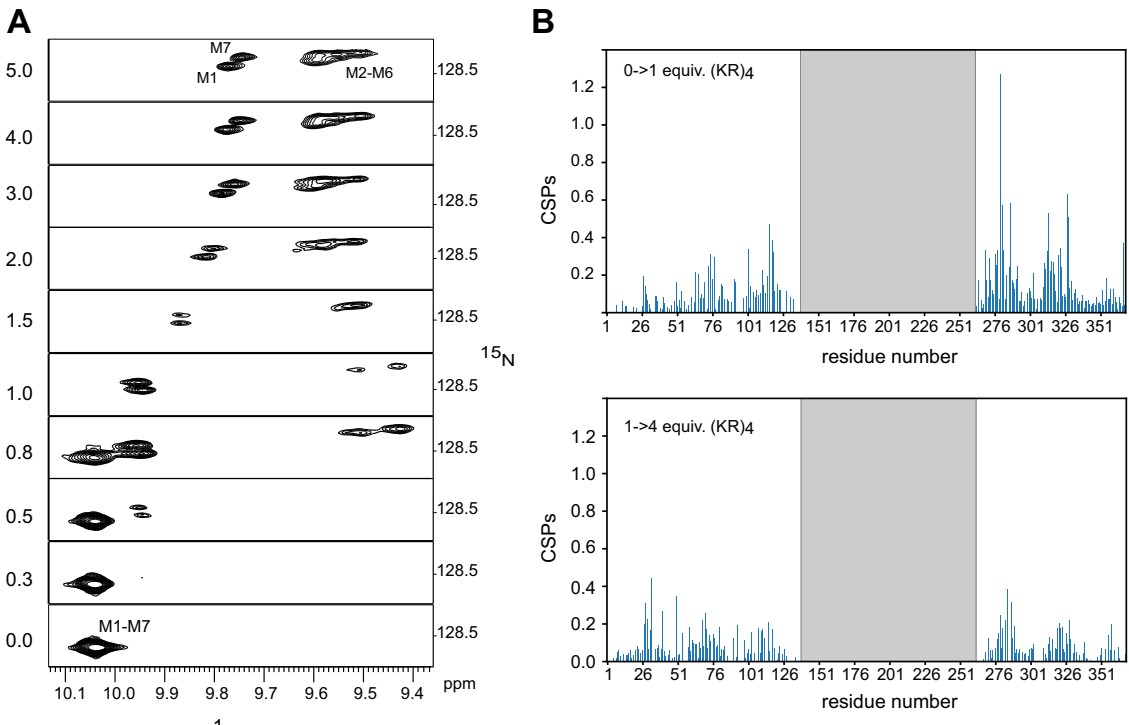

**Fig. 6 | Chemical shift changes of $N^YM_7C$ upon addition of $(KR)_4$. A** Expansion of the indole region of [$^{15}$N,$^1$H]-HSQC spectra. Added equivalents of peptide are depicted on the left. **B** CSP data when adding 1 (top) or 4 (bottom) equivalents of $(KR)_4$. The gray shaded central area represents residues that were unlabeled in all constructs and therefore remained unassigned.

the first equivalent of peptide in C-terminally labeled segment compared to those in the N-terminal one. Upon further addition of peptide, CSPs are generally smaller and more evenly distributed. These data suggest that the first peptide molecule indeed has a slight preference for the C-terminal part of the protein.

## Discussion

In creating a modular systems of binding proteins[29,30], it is important to understand the dynamics of binding of longer peptides. In Armadillo Repeat Proteins, the binding is antiparallel and colinear, with one repeat binding 2 adjacent amino acids. During the binding process, incoming ligands initially form an encounter complex, in which the ligand is proximal to the protein but where the native contacts have not been correctly formed yet[31,32]. However, not every encounter between ligand and protein results in successful formation of the protein-ligand complex because orientation of the ligand is important[33-35]. Most ligands and their corresponding binding pockets are highly asymmetric, and hence only a small fraction of all collisions results in productive formation of the protein-ligand complex, even if approached from the correct side. Encounter complexes with wrong ligand orientations usually cannot rearrange into the correctly bound state but require complete unbinding and correct rebinding. Here, we have investigated binding of peptides with unusually high affinities ($K_d$ approx. $10^{-13}$ M) to repeat proteins that, due to their modular nature, can be easily modified to expand the interaction surface. The high repetitiveness in both protein as well as peptide sequences in the studied system allows for more encounters to become productive. We also suggest that non-canonical binding modes can isomerize into the correct complexes without unbinding (vide infra).

Figure 7 summarizes the basic steps that might occur during binding of a (KR)-type peptide to the dArmRP with identical internal repeats. After the initial encounter of protein and peptide, various states II-IV can be formed that do not represent the lowest-energy complex, as well as the lowest energy state V. One of those states represents the canonical binding mode in which, however, not all contacts are properly made yet (II), because the geometry of the protein is slightly different from what would be perfect. Alternatively, a

register shift can occur during binding (III). Such a behavior is possible due to the highly repetitive nature of both ligand and receptor in this system. An alanine scan of the peptide revealed that Lys and Arg contribute differently to the binding energy, but their combined contribution is independent of their position in the primary sequence, and equals roughly 11.2 kJ/mol per KR dipeptide unit[9]. Off-rates for binding modes with 1 or 2 register shifts may therefore still be fairly slow. Finally, instead of the entire peptide being register-shifted, a few residues might simply loop out such that again less than the maximum number of interactions are made. Such a scenario is also rather likely because the incoming peptide is unstructured and because the formed contacts again are sufficient for fairly tight binding. All of the non-canonical states II-IV can isomerize to the canonical state V, with maximal interactions between peptide and protein. It should be noted that previous single-molecule experiments with the same system in $NM_5C:(KR)_5$ length showed that the majority of molecules would assume the canonical state V, but nevertheless a sizable fraction at equilibrium appeared to be register-shifted[28]. These experiments did not give any information on kinetics, however.

Surprisingly, we did not observe changes in spectra over time and therefore assume that initial equilibration to the canonical state is rapid compared to the times required to record the 2D spectra. We thus conclude that the system equilibrates into the canonical form much more rapidly than anticipated. So, why is equilibration fast, although states in which 6 out of the 7 possible contacts are formed are expected to be fairly long lived? We believe that two features are important for the equilibration process in that respect: Firstly, the fact that both ligand and receptor are repetitive so that shifting the entire peptide (or parts thereof) by one unit is possible by a reptation motion, allowing the same contacts to be formed. Secondly, there is a substantial electrostatic component in the binding which allows (i) keeping the peptide in the vicinity to the protein even when distances in the contacts are stretched or contacts even transiently broken and (ii) displacing contacts from one residue by those from another one without requiring close contact. Displacing one contact with another could trigger a cascade of events that will lead to the canonical state (insert b in Fig. 7). Note that this does not require breaking of many

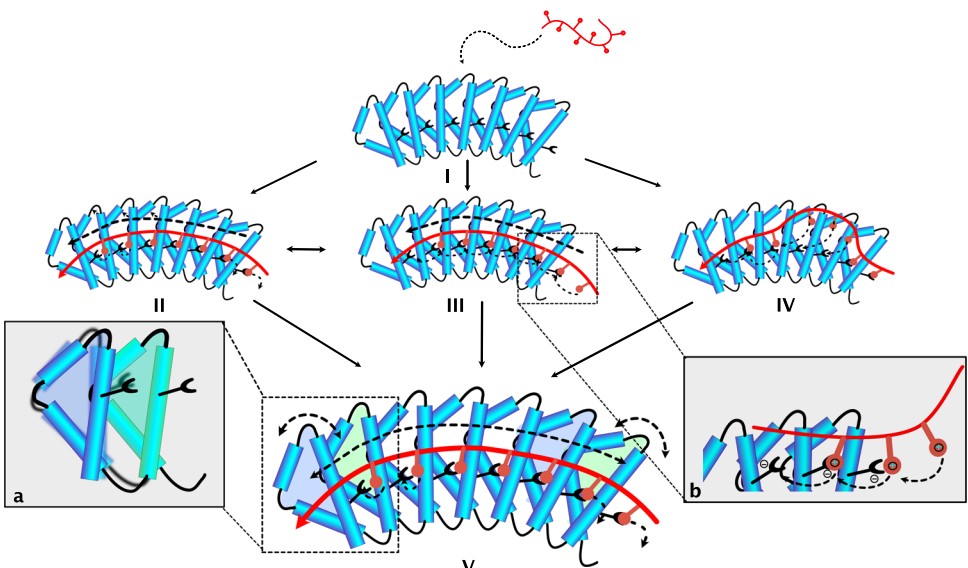

**Fig. 7 | (KR)$_n$ -type peptide binding to dArmRPs.** When the peptide binds to the apo dArmR (**I**) it can associate such that not all distances over the binding epitope match perfectly (**II**), it can bind with a register shift (**III**) or such that one or more (KR) units are looped out and form no contacts with the protein (**IV**), still retaining sufficient binding energy to not dissociate. These states can then equilibrate towards the canonical binding mode by alterations of the supercoil and transient

breakage of contacts (**V**). The insert (**a**) indicates the type of motion that will alter the supercoil, insert (**b**) the displacement of contacts (see text). Helices are depicted as blue cylinders, the peptide in red, and interacting residues as red spheres or black bowls. Motions in the peptide or the protein are indicated by dashed arrows.

## Table 1 | Binding of (KR)$_{4-7}$ to N$^Y$M$_7$C

| peptide | $k_{on}$ (M$^{-1}$ s$^{-1}$) | $k_{off}$ (s$^{-1}$) | $K_d$ (M) |
|---|---|---|---|
| (KR)$_4$ | $(2.9 \pm 0.12) \cdot 10^8$ | $(2.3 \pm 0.0016)^{-1}$ | $7.93 \cdot 10^{-10}$ |
| (KR)$_5$ | $(3.5 \pm 0.15) \cdot 10^8$ | $(1.7 \pm 0.029) \cdot 10^{-2}$ | $4.86 \cdot 10^{-11}$ |
| (KR)$_6$ | $(6.3 \pm 0.19) \cdot 10^8$ | $(3.0 \pm 0.098) \cdot 10^{-3}$ | $4.76 \cdot 10^{-12}$ |
| (KR)$_7$ | $(6.2 \pm 0.20) \cdot 10^8$ | $(3.0 \pm 0.13) \cdot 10^{-4}$ | $4.84 \cdot 10^{-13}$ |

Kinetic binding parameters of peptides of increasing length to a dArmRP with seven internal repeats, represented as mean ± S.D.

contacts simultaneously, an event that is very unlikely and hence would be slow. Also note that this process would allow more encounter complexes to result in productive canonical binding.

Stopped-flow measurements demonstrated that the $K_d$ for the longest peptide studied here, (KR)$_7$, is in the femtomolar range[9] (Table 1). Such an affinity would clearly place the off-rate and hence exchange processes into the slow regime with absolutely no expected contributions to the linewidth from exchange processes. One important observation of the present study, however, was that even the femtomolar binders display dynamics in their canonically bound state. Apparently, exchange processes on the ms time scale are present even in the tightest binding dArmRPs. The off-rates required to see this behavior for a two-state exchange process would be far too high in comparison to the actual rates based on the known $K_d$'s, $k_{on}$ and $k_{off}$ from stopped-flow data (Table 1).

To properly describe the observed line-shapes of dArmRP peaks during the peptide titrations a three-state process is thus required— one step characterizing overall binding/unbinding with very fast association rate constant ($>10^8$ M$^{-1}$ s$^{-1}$) and low dissociation rate constant (as low as $3 \cdot 10^{-4}$ s$^{-1}$) (Table 1) and a second step of first order with rate constants of $k_2 = 30$ s$^{-1}$ and $k_{2'} = 700$ s$^{-1}$ (Fig. 5) that is responsible for the observed line-broadening. Since the initial complex formation rate is governed by the sum of association and dissociation rates, it will equilibrate in a fraction of a second with micromolar protein and peptide. This second equilibration to the canonical complex must be due to transient breakages of individual contacts but unrelated to complete unbinding. The transient breakages of individual contacts may be related to the structural adaptations stemming from a slight mismatch between optimal distances in the receptor and peptide (e.g. state II in Fig. 7). They may, however, also be simply a property of the canonically bound state that would also take place if these distances would match perfectly: an individual side chain may temporarily leave its binding pocket and return, as it is still constrained by the rest of the bound peptide. It is interesting to note that in crystal structures of dArmRPs with (KR)-type peptides not all Arg residues were positioned in their canonical binding pockets[9].

KR-type peptides are unstructured in their unligated state, the association with the protein freezes many rotatable bonds both in the backbone as in the sidechains, and hence binding is expected to come with a big loss in entropy. Drug-receptor interactions are therefore usually limited to $K_d$s higher than 10 pM[36] since increasing binding enthalpy is often canceled by a decrease of the resulting entropy of the system (enthalpy-entropy compensation)[5,36–38]. Transient disruptions of contacts, while the ligand is remaining tightly bound, could potentially reduce the entropic loss and therefore result in higher affinities, which might seem counter-intuitive. Purely enthalpic binding might not fully exploit the possibilities of maximizing the free energy of binding, however, current design principles largely ignore residual entropy after binding. The proposed concept may equally apply to any larger peptide that is flexible in the unliganded state, especially with backbone and side chains bound by multiple defined interactions as in the Armadillo system described here. However, it likely requires

electrostatic components to ensure efficient rebinding and a larger number of such interactions so that the loss of a few interactions does not result in immediate unbinding. Importantly, each pocket for each side chain must still be very specific for this strategy to lead to both very tight and very specific binding. The principle might also be applicable to non-repeat proteins, provided the individual (un)binding of residues/moieties is possible – the increased number of productive encounter complexes and the fast equilibration dynamics towards the lowest energy complex, however, will only be observed for repeat proteins in combination with repetitive ligand sequences. The interface does not need to have a high degree of disorder but the side chains must be able to bind and unbind individually. It likely also poses restrictions on the nature of the binding interface of the protein as it should rather not constitute a deep cleft to provide sufficient space for the ligand dynamics without the requirement for major structural reorganizations. The principle of interaction would not be expected to change with length. Nonetheless, the total binding free energy cannot increase forever but would be intuitively expected to level off. Presumably, for very long systems the peptide would not behave as a single unit but as linked shorter peptides.

If individual contacts rapidly form and break, the structural changes required for binding may also be limited since not all contacts must be formed simultaneously. Using solution NMR techniques, we observed here that binding of (KR)$_4$ to N$^A$M$_4$C results in a gentle regularization of the binding surface, where the strong supercoil for modules is significantly reduced at both termini, although not to the extent observed in crystallographic studies. In crystal structures of ArmRP complexes with (KR)-type peptides the overall protein geometry is highly conserved[9,17,21], and much more regular than in the solution structure of the (KR)$_4$:N$^A$M$_4$C complex. We detected fairly high C$\alpha$(P/P + 2) distances, averaging around 8.2–8.6 Å in solution, as opposed to $7.35 \pm 0.42$ Å and $7.03 \pm 0.21$ Å (mean ± S.D.) found in crystal structures for the apo and canonically bound states[20]. We suspect that the larger variability in the solution state is due to dynamic fluctuations of individual contacts that reduces the need for a highly regular binding interface required to form all contacts simultaneously. In that context it is interesting to note that natural Armadillo repeat proteins (nArmRPs) also do not generally possess a curvature ideal for extended modular binding, displaying C$\alpha$(P/P + 2) distances of about 8.5 Å[8].

We used NMR to specifically look at changes in conformational entropy, as calorimetric methods such as isothermal titration calorimetry (ITC) report only on overall changes in entropy. In fact, it is almost impossible to measure the conformational entropy in such systems with calorimetric methods because of the unknown size of the entropic contribution from the solvent. Moreover, classical techniques to determine dissociation constants such as surface plasmon resonance (SPR), ITC or fluorescence polarization (FP) techniques are largely insensitive to partial unbinding. Nonetheless, the structural details of protein-peptide interactions are mostly determined from high-resolution crystal structures. We realized, however, that X-ray techniques can only give limited information, first because subtle structural details such as the supercoil of the ArmRPs are likely influenced by crystal packing forces[9,20–23,39]. Most importantly, as shown in this work and by others, solution NMR potentially presents a method that can investigate the binding dynamics in the bound state at atomic resolution and monitor the protein response in solution.

Using these approaches, we learned that very tight binding can be achieved, if residual ligand entropy remains present in the bound state, such that the binding enthalpy is maximized without paying the full cost of entropic loss, since the ligand is never completely frozen. We may have uncovered a new principle of very tight binding that could guide design principles for ultra-tight binders and may also suggest to revisit properties of known tight binders.

# Methods

## dArmRPs nomenclature

In this paper we introduce a simplified dArmRPs nomenclature. Each dArmRP construct is composed of an N-terminal cap (N) followed by a variable number of identical internal modules (M) and completed by a C-terminal cap (C). Superscripts are used to discriminate between different design versions of the same component, such as $N^Y$ and $N^A$, while subscripts are used to indicate multiple repetitions, such as $M_4$. Internal modules followed by an in-line number mark the ordinal number counting from the N-terminal module, i.e., M3 is the third internal module. Comparing to previous publications on dArmRPs, the C-terminal cap indicated with "C" in this publication is identical to the A cap, and the N-terminal $N^Y$ cap is identical to the Y cap in previous publications.

## Cloning, expression and purification

Genes for $N^Y M_4 C$ and $N^A M_4 C$ were cloned into the pEM3BT2 vector[40] containing a TEV-cleavable N-terminal $(His)_6$-GB1 domain[41] using XbaI and BamHI restriction sites. Cys mutants required for site-specific spin labeling were produced through the QuikChange II mutagenesis protocol (Stratagene), utilizing primers purchased from Microsynth. Genes for $N^Y M_7 C$ and $N^A M_7 C$ were purchased from Genscript.

Uniformly labeled proteins were expressed in *E. coli* BL21 (DE3) cells at 37 °C in 500 mL M9 medium[42], containing either 3 g/L of $^{13}C$ glucose and 1 g/L $^{15}N$ $NH_4Cl$, or the unlabeled equivalents and induced with 1 mM IPTG at an OD600 of ca. 0.6 for 16 h at 30 °C. After harvesting by centrifugation, the obtained cell pellet was re-suspended in 15 mL buffer A (50 mM sodium phosphate at pH 7.7, 500 mM sodium chloride, 20 mM imidazole, 30 μM sodium azide) on ice. The re-suspended cells were disrupted in a single passage through a French Press (Thermo Electron Corporation) at 1100 psi pressure and 4 °C, and the obtained lysate was mixed with ca. 1 mg of DNaseI (Roche, Switzerland) and cleared by centrifugation for 30 min at $30,000 \times g$ and 4 °C. The supernatant was filtered through a Filtropur S 0.2 μm sterile filter (Sarstedt, Germany) and was passed over a 5 mL HisTrap HP column (GE Healthcare) in buffer A. After washing with 15 column volumes of buffer A, the target proteins were eluted in a linear gradient of 20–500 mM imidazole in 100 mL buffer A. The fractions containing the desired target protein were then mixed with 2 mg TEV protease[41] and dialyzed overnight at room temperature (RT) in a 3.5 kDa MWCO dialysis membrane against 2 L TEV cleavage buffer (50 mM sodium phosphate at pH 7.7, 100 mM sodium chloride, 0.5 mM DTT, 25 μM EDTA and 30 μM sodium azide). The proteolytically cleaved protein solution was then again passed over the 5 mL HisTrap HP column in buffer A and the eluate containing the desired target protein devoid of the N-terminal $(His)_6$-GB1 domain was collected for two consecutive dialysis steps at RT in a 3.5 kDa MWCO dialysis membrane (Carl Roth, Switzerland), each for 8 h against 2 L of fresh NMR buffer (20 mM sodium phosphate at pH 7.0, 50 mM sodium chloride, 30 μM sodium azide). The protein solution was then concentrated in a 3 kDa MWCO Amicon Ultra-15 centrifugal filter devices (Millipore) at $3500 \times g$ and 16 °C until the desired concentration was obtained. In order to remove co-purifying contaminants, uniformly labeled $N^Y M_7 C$ and $N^A M_7 C$ required an additional purification step by performing a full denaturation in 8 M urea prior to Ni-NTA chromatography and then refolding the protein on-column by exchanging the buffer with a gradient toward the absence of urea.

Constructs required for segmental labeling were cloned into the pEM3BT2 vector[40]. N-terminal fragments were flanked by a $(His)_6$-GB1-3C site element at the 5′-end and by the intein MxeGyrA at the 3′-end. C-terminal fragments were fused to a $(His)_6$-GB1-TEV site at the 5′-end, which quantitatively produces the N-terminal Cys, required for expressed protein ligation (EPL), through the TEV recognition site ENLYFQ/C. The Cys mutation was introduced by replacing Ser-21 in internal modules, located in the H2-H3 loop. Five segmentally labeled constructs were prepared according to the following scheme, where

the letter in parenthesis indicates whether the N- or C-terminal fragment was isotope-labeled: $N^Y M_4 C$ (N) = $N^Y Mab + cM_2C$; $N^Y M_4 C$ (N) = $N^Y MMab + cMC$; $N^Y M_4 C$ (C) = $N^Y MMab + cMC$; $N^Y M_7 C$ (N) = $N^Y MMab + cM_4 C$; $N^Y M_7 C$ (C) = $N^Y M_5 ab + cMC$ (Supplementary Fig. 1). Coupled N- and C-terminal fragments forming the full protein were expressed in parallel in *E. coli* BL21 (DE3) cells (Stratagene) according to the protocol described above for uniformly labeled proteins. After purification of both fragments by Ni-NTA chromatography, the His-tag of the C-terminal fragment was removed by TEV cleavage and subsequent reverse Ni-NTA chromatography.

The two fragments were fused together by expressed protein ligation (EPL)[43, 44] by mixing them in a 1:1 molar ratio in EPL buffer (50 mM TrisHCl, 500 mM NaCl, 0.1 mM TCEP, 0.1 mM EDTA, 8 M urea, pH 8) in presence of 100 mM MESNA (2-mercaptoethanesulfonate) at RT for 24 hours. MESNA was then removed by dialysis and the fused construct separated from the liberated MxeGyrA under denaturing conditions in 8 M urea by Ni-NTA chromatography. Subsequently, the protein was refolded on column through a buffer exchange gradient to remove urea, while unligated N-terminal fragment was removed by preparative SEC. 3 C cleavage was used to remove the N-terminal His tag, followed by reverse Ni-NTA chromatography to yield the desired pure ligation product. Final NMR samples at 100-200 μM were prepared by exchange to the NMR buffer (20 mM $Na_2HPO_4$, 150 mM NaCl, 10% $D_2O$, pH 7.0).

While expression yields of full-length ArmRP proteins were typically very high (up to 200 mg/L), yields of the intein fusions and the (shorter) C-terminal fragments were lower, and additionally compounded by problems related to protein stability. In particular, the sensitivity of proteins to degradation due to instability of the N cap led to the development of proteins with an optimized N cap (e.g. $N^A M_4 C$). Yields for expressed protein ligations were variable (20-60 mg/L, depending on constructs) but around 30% for constructs with optimized Cys positions.

Purification of $N^Y M_4 C$ was straightforward and required a single Ni-NTA purification step. In case of $N^Y M_7 C$ we noticed co-purifying contaminations. To remove them, the protein was denatured in urea prior to the chromatographic step, and refolded after chromatography by dialysis against the NMR buffer. Even when using triple labeling ($^2H$, $^{13}C$, $^{15}N$) for the labeled part, using this protocol a pure 300 μM NMR sample of segmentally-labeled protein in 220 μL was produced from 1 L of bacterial culture.

## Site-specific spin labeling

Side chains of introduced unique Cys residues were used for the attachment of the Tm-3R4S-DOTA-M7-Thiazole tag, in order to produce the paramagnetic proteins or their corresponding diamagnetic references according to a published protocol[45]. In brief, proteins (typically 150 μM, 500 μL) were reduced in tagging buffer (20 mM $Na_2HPO_4$, 0.2 mM TCEP, pH 7.0) for 10 min, and then a PD-10 column (Sigma) was used to separate the protein solution from the reducing agent. A fivefold excess of lanthanide tag was incubated overnight with the protein at room temperature. Final NMR samples (150 μM) were prepared by exchange to an NMR buffer (20 mM $Na_2HPO_4$, 10% $D_2O$, pH 7.0), and this buffer exchange also served to remove excess unreacted tag.

## Measurements of kinetic parameters using a FRET based stopped-flow assay

Kinetic binding constants were measured using the FRET-based assay developed by Ernst et al.[28] in a PiStar-180 stopped-flow fluorometer (Applied Photophysics Ltd.) equipped with a mercury–xenon lamp. sfGFP was excited using a wavelength of 436 nm with a bandwidth of 10 nm[28] and fluorescence was collected using a 590 nm long-pass filter. A constant concentration (20 nM) of GFP-tagged peptides $((KR)_n$ with $n = 4$–7) was mixed with increasing concentrations (50 nM, 100 nM,

125 nM, 160 nM, 250 nM) of mCherry-tagged $NM_7C$ in a 1:1 volume ratio in a 120 μL internal mixing and observation chamber in the instrument to determine $k_{on}$. $k_{off}$ was determined from a competition assay, in which a preincubated complex of 50 nM tagged peptide and 40 nM tagged protein were mixed with 2 μM of unlabeled competitor in a 1:1 ratio.

## NMR measurements

All experiments were recorded on Bruker Neo 600 or 700 MHz spectrometers, employing cryogenically cooled or Prodigy $^1H$, $^{13}C$, $^{15}N$ triple-resonance probes. Pulse field gradients with coherence selection[46] were applied to all heteronuclear $^{15}N,^1H$ experiments using standard Bruker pulse sequences, together with the Rance-Palmer method for sensitivity enhancement[47]. All spectra were processed in TOPSPIN using cosine-bell-shifted window functions prior to Fourier transformation, and analyzed in CARA[48]. Chemical shift assignments were obtained from triple-resonance and HCCH-type 3D NMR experiments using the standard Bruker pulse-program library. Assignments of $NM_7C$-type proteins were performed using samples that were additionally perdeuterated and experiments that use $^2H$ decoupling when required.

In order to obtain spectra required for the determination of PCS employed in the refinement, proteins coupled to the dia- or paramagnetic tag were subjected to buffer-exchange to NMR buffer (20 mM $Na_2HPO_4$, 2 mM trimethylsilylpropanoate (TMSP), pH 7.0 and 10% $D_2O$) using centrifugal filters (Amicon) to remove any unreacted lanthanide tag. $[^{15}N,^1H]$-HSQC spectra were then recorded at 293 K with spectral widths of 15 and 40 ppm in the direct and indirect dimensions, respectively, using 1024 or 128 complex data points. Paramagnetic states were further analyzed by 200 ms NOESY-$[^{15}N,^1H]$-HSQC experiments to determine amide-amide NOEs. Proteins were in the 100 – 150 μM or 350 – 400 μM concentration ranges for tagged and untagged samples, respectively. TMSP was used as internal reference to calibrate the proton chemical shift, from which the nitrogen chemical shift was indirectly referenced to the liquid ammonia scale using the conversion factor of 0.10132900[40].

During titrations with the peptides, the $(KR)_{4/5/6/7}$ peptides were purchased from Genscript and produced by chemical synthesis with an acetylation at the N-terminus and no modifications at the C-terminus and using TFA counter ions. The peptides were dissolved in NMR buffer to generate 10 mM stocks that were titrated in 0.1 molar equivalents steps to 250 μM dArmRPs in NMR buffer up to 1.5 equivalents, and then in 0.5 equivalents steps up to 5.5 equivalents. For each step, a $[^{15}N,^1H]$-HSQC experiment was recorded at 310 K with the other parameters described above.

## Lineshape simulations and extraction of kinetic parameter

The change of line-shape and peak positions in the $[^{15}N,^1H]$-HSQC spectra upon adding $(KR)_n$ was simulated using in-house written Mathematica notebooks. In principle, for the simple two-state process of ligand binding, described only by single $k_{on}$ and $k_{off}$ rates and the concentrations used, the modified Bloch McConnell equations[49] can be used. However, to be able to model more complex models of binding, the simulated line-shapes were obtained by numerical integration of the corresponding differential equations describing the exchange of magnetization. The line-shapes of the experimental data were obtained by summing up F2-slices of the $[^{15}N,^1H]$-HSQC spectra by taking the F1-projection covering the individual peaks. The such-obtained 1D-slices were then compared against the simulated spectra, and the parameters (in particular: rate constants) adjusted to minimize the difference between experimental and simulated data. Parameters were changed to optimize agreement of peak position and line-widths.

For the simulation of line-shapes and peak positions in the $[^{15}N,^1H]$-HSQC spectra during the titration with peptides an additional exchange process for the bound state was modeled according to Eq.

(1). In that equation $k_{1,on}$ and $k_{1,off}$ are the on- and off-rates describing the canonical binding of the peptide, whereas $k_2$ and $k_{2'}$ are the rate constants for a unimolecular "isomerization" process, i.e., the second exchange process involving transient breakages of individual contacts.

Transverse magnetization of free ($M_P$), bound ($M_{PL}$) or partially (transiently) unbound protein ($M_{PL^*}$) is then described by the following system of coupled differential equations:

$$\frac{dM_P(t)}{dt} = (-i\Omega_P - R_2 - k_{1,on}[L])M_P(t) + k_{1,off}M_{PL}(t) \qquad (2)$$

$$\frac{dM_{PL}(t)}{dt} = (-i\Omega_{PL} - R_2 - k_{1,off} - k_2 M)_{PL}(t) + k_{1,on}[L]M_P(t) + k_{2'}M_{PL^*}(t) \qquad (3)$$

$$\frac{dM_{PL^*}(t)}{dt} = (-i\Omega_P - R_2 - k_{2'}M)_{PL^*}(t) + k_2 M_{PL}(t) \qquad (4)$$

For simplicity, the same transverse relaxation rate $R_2$ was assumed for all protein states and the chemical shift of the partially unbound state $PL^*$ to be the same as for the free protein, that is $\Omega_P$. $\Omega_{PL}$ is the chemical shift of the bound state and $[L]$ the equilibrium concentration of the free peptide. Simulated data are obtained from numerical integration of the system using the NDSolve function within Mathematica and starting from transverse magnetization given by the proportion of equilibrium concentrations:

$$[P] = [P_0] - [PL]A \qquad (5)$$

$$[PL^*] = \frac{k_2}{k_{2'}}[PL] \qquad (6)$$

$$[L] = [L_0] - [PL]A \qquad (7)$$

with $[PL]$ given by the solution of

$$k_{1,off}[PL] - ([L_0] - A[PL])([P_0] - A[PL])k_{1,on} = 0 \qquad (8)$$

and $A = (1 + \frac{k_2}{k_{2'}})$. The so obtained time-domain data is then Fourier-transformed and compared to the experimental data and the parameters $R_2$, $\Omega_P$, $\Omega_{PL}$, $k_{1,on}$, $k_{1,off}$, $k_2$, $k_{2'}$ and an overall scaling factor were adjusted to minimize the difference between simulated and experimental data (judged from visual inspection). Thereby the $k_{1,on}$ and $k_{1,off}$ rate constants were chosen such that their ratio was in the range of the measured or predicted $K_d$ values[21] and that all signals during the titration could be fitted well (see text, Supplementary Fig. 12). The decay of magnetization due to relaxation and exchange-broadening during the INEPT sequences was taken into account by starting the simulation with a spin-echo sequence of corresponding echo-time (11 ms).

## Chemical shift perturbation (CSPs) determinations

Assignments of the spectra required for CSPs were performed through segmental labeling as detailed in the Results section. Chemical shifts of $N^Y M_4 C$, $N^A M_4 C$ and $N^Y M_7 C$ in the $(KR)_4$-bound state were extracted at 2 molar equivalents of the peptide. CSPs were calculated using the following formula[50]

$$\sqrt{\frac{(\delta H_f - \delta H_b)^2}{1} + \frac{(\delta N_f - \delta N_b)^2}{4}} \qquad (9)$$

where $\delta H$ and $\delta N$ represent the chemical shifts in the proton and nitrogen dimensions, respectively, while $f$ and $b$ represent the free and

bound states, respectively. Unassigned residues (in either state) were set to zero CSP.

## Structure refinements

The Python module Paramagpy[51] was used to calculate the paramagnetic anisotropic susceptibility tensors and the corresponding back-calculated PCS together with Q-factors (Supplementary Figs 5–9, Supplementary Data 1). Structure refinements were performed in an iterative procedure using simulated annealing in CYANA[52] as previously described[19]. Four initial models, used as starting structures for the PCS-driven refinement, were prepared by adapting the models described[19] by mutating the $N^Y$ cap into the $N^A$ cap using the PyMOL mutagenesis wizard. Those models were subsequently refined in ten cycles, each determining 500 conformers in 25,000 MD steps with pseudocontact shift (PCS) weight of 50 and upper distance limit (UPL) weight of 1. In order to increase the statistical significance of our results, we applied the refinement protocol to four different starting models, applying PCS for either the apo or bound state. The resulting convergence, defined as the RMSD between resulting structures at each cycle, reaches $0.44 \pm 0.03$ Å and $0.56 \pm 0.03$ Å (mean ± S.D.) for the apo and bound state, respectively (Supplementary Fig. 6), after 10 cycles. Paramagpy Q-factors of paramagnetic anisotropic susceptibility tensors ($\Delta\chi$-tensors) also improve steadily throughout the iterations (Supplementary Fig. 7), and validation with different seeds displayed good convergence with an average RMSD of $0.39 \pm 0.24$ Å for the apo state and $0.27 \pm 0.07$ Å for the bound state (Supplementary Table 1), indicating that the refinement protocol is correct. Note that no corrections for residual anisotropic chemical shifts (RACS) were done, similarly as in the originally published protocol[19]. However, we eliminated $^{15}N$ PCS derived from peaks where the line connecting diamagnetic and paramagnetic species has a slope outside the range of 0.8 - 1.2 to remove exceptionally large RACS that would have created a local artifact in the calculation. PyMOL was used for visual structural comparison, and RMSD calculations were performed considering only backbone atoms either with CYANA, the PyMOL *align* function or custom Python scripts.

## Curvature analysis

The supercoil was determined through measurements of $C\alpha(P/P + 2)$ distances in the Rosetta symmetry framework, as previously detailed[8] and illustrated in Fig. 3B. In brief, Rosetta was used to generate models with uniform curvatures of two neighboring internal modules by extending the two modules in question to a regular nine-mer. A second model consisting of a single dArmRP internal module bound to a (KR)-dipeptide with an ideal geometry was then superimposed twice to adjacent modules of the nine-mer. This allowed the calculation of the $C\alpha(P/P + 2)$ distance, which refers to the distance between the $C\alpha$ of two Arg in the peptide.

## Reporting summary

Further information on research design is available in the Nature Portfolio Reporting Summary linked to this article.

## Data availability

Coordinates of unliganded (generated in a previous study[14]) and $(KR)_4$-bound $N^AM_4C$ (generated in this study) were deposited in the PDB database under accession codes 7R0R [https://doi.org/10.2210/pdb7R0R/pdb] and 8OH7 [https://doi.org/10.2210/pdb8OH7/pdb], respectively. Chemical shifts for the diamagnetic references were deposited for the complex with $(KR)_4$ on the BMRB database under codes 51290, 51291, and 51292 for the three different attachment sites of Tm-tagged proteins. The raw data for Figs. 1–6 and Supplementary Figs. 1-16 are provided as a Source Data file. All data used during the PCS-restrained refinements have been deposited in the database Zenodo [https://doi.org/10.5281/zenodo.8435468]. Source data are provided with this paper.

## Code availability

A tutorial containing a protocol capture for the PCS refinement with the relevant Python scripts and Cyana macros is available on Github [https://github.com/Evets90/Iterative_PCS_refinement]. The code has been generated as part of a previous study[19] but used in this study to compute the structure PDB 8OH7. Further Python scripts and CYANA macros are available upon request.

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

## Acknowledgements

We acknowledge Ben Schuler and Mate Erdelyi for critical reading of the manuscript. This work was rooted in a project financed via a SINERGIA grant from the Swiss National Science Foundation (funder id: 10.13039/501100001711, grant no. 122686 to O.Z., A.P.) and the European Commission (FET-Open Grant 764434 PRe-ART, A.P.)

## Author contributions

Conceptualization: O.Z., S.C.; Data curation: S.C., M.S., E.M., P.E.; Formal analysis: S.C., S.J.; Funding acquisition: O.Z., A.P.; Investigation: S.C., M.S., E.M., W.D.; Methodology: S.C., D.H., S.J., O.Z.; Project administration: O.Z.; Resources: D.H., T.M.; Software: S.C., S.J.; Supervision: O.Z., A.P.; Validation: S.C., D.H., A.P.; Visualization: S.C., M.S., S.J., O.Z.; Writing – original draft: S.C., O.Z. Writing – review & editing: S.C., O.Z., A.P., D.H., S.J., E.M., M.S.

## Competing interests

The authors declare no competing interests.
