## [Peer Review File · Nature Communications]

REVIEWER COMMENTS

Reviewer #1 (Remarks to the Author):

The article describes exchange broadened cross-peaks in HSQC spectra of an armadillo repeat protein binding various KR repeat peptides. The authors attribute the line broadening occurring despite very tight binding to chemical exchange in the peptides, which can switch between different electrostatic interaction partners without dissociating from the protein. This provides an entropic contribution favouring complex formation. The experimental evidence is convincing, and the insight gained worth publication.

Minor points:

Page 5: 'following reversely a ligand titration' is awkward and unclear wording.

In the top panel of Fig. 1c, is one of the peaks labeled '3b'?

Page 7: which construct is NAM4C4?

Fig. 2: which colour indicates weak perturbation?

Fig. 3: the $\text{C}\alpha(\text{P}/\text{P}+2)$ distance is confusing nomenclature – it is measured between two atoms of the peptide ligand, but apparently also measurable in the apo-protein, i.e., in the absence of ligand. It would help the reader to explain this nomenclature here (although apparently the term is established in the field).

Reference 14 is missing page numbers, reference 44 is missing details.

Fig. S5: please indicate the ppm values of the isosurfaces shown.

Table S4: the title is unclear – what is '10 tg v'? Furthermore, what are the units of the tensors?

The authors report PCS values for ^{15}N spins. Were they used for structure refinement? If so, were residual CSA effects taken into account? These may be significant, as the tensors were quite large (John et al., JACS 127, 17190, 2005).

Segmental isotope labeling is not trivial, and it would therefore be interesting to have some information about the protein expression and ligation yields.

Where am I supposed to find the references cited in the supplementary material?

Reviewer #2 (Remarks to the Author):

In this interesting manuscript the authors describe and analyse an unexpected finding for high flexibility in ultrastrong protein peptide complexes. The authors deduce their findings from the careful inspection of line widths which indicate transient breakages of electrostatic interactions at certain locations in the complex, with much faster off rates than expected for ultrastrong binding. These complex dynamics are suggested to compensate for entropy losses in the flexible peptide which is usually thought to be frozen in the optimal distance of all binding sites.

For their systematic investigation the authors use designed Armadillo repeat proteins dArmRP and complementary (KR)_n peptides with matching and non-matching numbers of repeats. The internal 3 helix modules each bind to a KR dyad by pi-cation interactions. The largest NM7C protein binds to the corresponding (KR)₇ peptide with femtomolar affinity, as determined by stopped flow experiments.

From peak shifts of critical glycine sensor protons in HSQC experiments the authors locate the binding sites and follow kinetics of peptide binding. Structural changes in the protein are taken from pseudocontact shifts (PCS).

The authors discovered unexpected line broadening even in the tightest complexes and postulate that a second relatively fast local exchange process within the formed protein peptide complex occurs. These exchange processes seem to involve only one or few unproductive bond breakages at the same time and do not significantly lower the overall affinity, albeit raise conformational entropy which is favourable to maintain ultratight binding.

This referee is not an expert in NMR spectroscopy and must acknowledge the concept of “pseudocontact shifts” as given. However, the findings and their interpretation make a lot of sense and may be potentially also transferred to other events of ultratight protein ligand binding. However, the specific example is limited to a highly symmetric, repetitive protein and similar peptide guests, which allow a substantial conformational freedom without giving up the most powerful noncovalent interactions that would result in immediate unbinding.

A drawback of this investigation is that only NMR spectroscopic details were used to underpin a far-reaching conclusion; although the authors argue, that ITC does not yield direct information on conformational entropy, because the determination of solvents entropy is difficult, I recommend to use this technique in order to find out the overall entropy component of the entire complex formation. Solvent entropies may be estimated from calculations, and thus help to decide, if the conformational loss is effectively counterbalanced due to the internal flexibility of the system.

A second independent experimental method is highly desirable in order to support the interesting and important hypothesis put forward by the authors. If this can be done, I recommend publication in Nature Communications.

Reviewer #3 (Remarks to the Author):

This paper is an elegant and comprehensive study of the underlying mechanisms of the high affinity for designed Arm repeats with unstructured short peptide ligands - a fascinating and remarkable system. It represents a substantial piece of work on a simple yet complex molecular interaction, with expressed protein ligation to make samples suitable for NMR, extensive dynamic analysis, and additional biophysical and in silico data. It was good to see the authors point out why one should not use ITC to draw conclusions about entropic effects, due to the unknown contributions from solvent, and therefore why NMR is such an important technique.

I recommend publication with two requests-

1. They write in the abstract: "The results of this work indicate novel design principles for ultra-tight binders" Please can the authors elaborate/comment on this how one might apply their finding to protein design.
2. The authors write: "Transient disruptions of contacts, while the ligand is remaining tightly bound, could potentially reduce the entropic loss and therefore result in higher affinities, which might seem counter-intuitive. This concept may equally apply to any larger peptide that is tightly binding to a protein, especially with backbone and side-chains bound by multiple defined interactions as in the Armadillo system described here. It is thus not expected to be limited to interactions with repeat proteins and peptides with repetitive sequences; however, it may require electrostatic components to ensure efficient rebinding. Importantly, it relies on a larger number of interactions such that the loss of a few interactions does not result in immediate unbinding." Can the authors speculate on how much they think this might be applicable to non-repeat proteins - they say it would need a large interface, but would that interface also need to have a high degree of intrinsic disorder? Also do they expect the effects observed to continue with increasing numbers of repeats, or is there a limit?

Reviewer 1:

Page 5: 'following reversely a ligand titration' is awkward and unclear wording.

We have rephrased that sentence to become: "Assignments were subsequently adapted to the apo state by tracing signals back from two to zero equivalents (Fig 1C)."

In the top panel of Fig. 1c, is one of the peaks labeled '3b'?

Yes, it should have been 3b. We are grateful to the reviewer for spotting it and we corrected the error.

Page 7: which construct is NAM4C4?

It is N^AM₄C. We corrected the typo.

Fig. 2: which colour indicates weak perturbation?

A color description was missing in the caption, it should be "weak - cyan (<0.25 and 0.11 ppm)"

Fig. 3: the C α (P/P+2) distance is confusing nomenclature – it is measured between two atoms of the peptide ligand, but apparently also measurable in the apo-protein, i.e., in the absence of ligand. It would help the reader to explain this nomenclature here (although apparently the term is established in the field).

The C α (P/P+2) value is used to evaluate the *curvature* of the construct rather than the *binding* directly. To this end an extended peptide is bound via 2 residues i and i+2 to an Armadillo peptide in the canonical mode such that the proper H-bonds are formed and the distance of C α atoms at positions i and i+2 is measured. This distance depends on the curvature of the ArmRP. The method is described in more detail in reference 8 (Reichen et al., JMB 2016) and now briefly described in the caption of Fig. 3 as well as referenced to the original paper by Reichen et al.

Reference 14 is missing page numbers, reference 44 is missing details.

We added the reference 44 and the details for ref. 14. Reference 44 is not a journal article but a manual in form of a book that describes the usage of the program CARA.

Fig. S5: please indicate the ppm values of the isosurfaces shown.

We have now added the ppm values for the depicted isosurfaces.

Table S4: the title is unclear – what is '10 tg v'? Furthermore, what are the units of the tensors?

We apologize for this mistake, and have now specified in the Table caption that these are the number of iterations from the PCS-restrained refinement protocol.

The authors report PCS values for ^{15}N spins. Were they used for structure refinement? If so, were residual CSA effects taken into account? These may be significant, as the tensors were quite large (John et al., JACS 127, 17190, 2005).

^{15}N PCS were used in addition to ^1H PCS data. The reviewer raises an important point. We have looked into this in the paper describing the development of the refinement protocol (J. Biomol. NMR 75 (2021), 319-334). At that time a similar question was asked by a reviewer and we replied:

“In order to investigate this effect we have performed simulations in presence of only ^1H PCS data, and the results are very comparable in case of the simulated PCS. We added the comparison as Fig. S17 to the Supp. Mat. (in that publication). In case of the experimental data, when excluding ^{15}N PCSs, computations result in an average variation of $\sim 0.85 \text{ \AA}$, once the unstable Y cap is removed from the calculation (page 16 top). The intrinsic accuracy threshold detected in the protocol is $\sim 0.5 \text{ \AA}$, so the result is within the expected range. In the end, adding ^{15}N PCS does not provide independent data since the ^{15}N nuclei are mostly close to the amide nitrogen when compared to the distance to the metal center.

In principle, peaks from diamagnetic and paramagnetic species are connected by lines that with a slope of about 1.0. In fact, we have eliminated all PCS data for which the connecting line possesses a slope greater than 1.2 or smaller than 0.8, because we suspected that those couples may actually present mis-assignments of the paramagnetic species. Thereby, we also might have eliminated PCS data that contain large artefacts from ^{15}N CSA.

Fig S17 of (J. Biomol. NMR 75 (2021), 319-334): Impact of backbone ^{15}N PCS in the iterative refinement protocol. **a** simulations using the optimized iterative protocol with virtual PCS. The accuracy of the calculation is plotted versus the cycle number (top) when the calculation is performed in presence (blue) or absence (red) of ^{15}N PCS. In all the simulations PCS values are computed from structure A (blue) but simulations start from different input structures, from left to right: structure B, structure C and structure D (see Materials and Methods). The resulting structure after 10 cycles of refinement is shown in red aligned with the target structure in blue (bottom). The RMSD between target and computed structure after superposition of backbone heavy atoms of the entire sequence is indicated below. In the top-right corner, the RMSD between all three input structures during the refinement (convergence) is depicted by a dashed blue (presence of ^{15}N PCS) or red (absence of ^{15}N PCS) line with the shaded region indicating the standard deviation. **b** refinement using the optimized iterative protocol with experimental PCS. The RMSD between all four input structures during the refinement (convergence) is depicted by a blue (presence of ^{15}N PCSs) or red (absence of ^{15}N PCSs) line, calculated when excluding the Y cap and the first internal module, with error bars indicating the standard deviation. On the bottom, the resulting structure after 10 cycles of refinement is shown in blue (presence of ^{15}N PCS) or red (absence of ^{15}N PCS). The backbone heavy atom RMSD between the two structures, excluding the Y cap and the first module, is displayed below each alignment. ”

In principle, the problem can be reduced by incorporating residual anisotropic chemical shifts (RACS). Since the method was developed and verified without using RACS, we prefer to stick to the original protocol.

Segmental isotope labeling is not trivial, and it would therefore be interesting to have some information about the protein expression and ligation yields.

ArmRP protein typically expresses at high yields. While expression yields of full-length $\text{N}^{\text{Y}}\text{M}_4\text{C}$ and $\text{N}^{\text{Y}}\text{M}_7\text{C}$ were typically very high (20-60 mg/L), yields for expression of the intein fusions and the (shorter) C-terminal fragments were lower, and additionally compounded by problems related to protein stability. In particular, the sensitivity of proteins to degradation due to instability of the N cap led to the development of proteins with optimized N cap (the proteins described in the paper as $\text{N}^{\text{A}}\text{M}_4\text{C}$). However, for the lack of time and resources we did not produce segmentally-labeled proteins with the new N cap.

Yields of ligation were a bit variable but up to 40 % for optimized Cys positions.

Purification of YM_4A was straightforward and only required a single Ni-NTA purification step. In case of YM_7A we noticed that contaminants were co-purifying from the affinity purification. To remove them, the protein was denatured in urea. Using this protocol we lost more than 50 % of protein during purification. We have added these details now to the methods section on page 19-20.

Where am I supposed to find the references cited in the supplementary material?

We apologize for the missing references for the SI. We have added them to the end of the revised version of the SI.

Reviewer 2:

In this interesting manuscript the authors describe and analyse an unexpected finding for high flexibility in ultrastrong protein peptide complexes. The authors deduce their findings from the careful inspection of line widths which indicate transient breakages of electrostatic interactions at certain locations in the complex, with much faster off rates than expected for ultrastrong binding. These complex dynamics are suggested to compensate for entropy losses in the flexible peptide which is usually thought to be frozen in the optimal distance of all binding sites.

For their systematic investigation the authors use designed Armadillo repeat proteins dArmRP and complementary (KR) $_n$ peptides with matching and non-matching numbers of repeats. The internal 3 helix modules each bind to a KR dyad by pi-cation interactions. The largest NM_7C protein binds to the corresponding (KR) $_7$ peptide with femtomolar affinity, as determined by stopped flow experiments.

From peak shifts of critical glycine sensor protons in HSQC experiments the authors locate the binding sites and follow kinetics of peptide binding. Structural changes in the protein are taken from pseudocontact shifts (PCS). The authors discovered unexpected line broadening even in the tightest complexes and postulate that a second relatively fast local exchange process within the formed protein peptide complex occurs. These exchange processes seem to involve only one or few unproductive bond breakages at the same time and do not significantly lower the overall affinity, albeit raise conformational entropy which is favourable to maintain

ultratight binding.

This referee is not an expert in NMR spectroscopy and must acknowledge the concept of “pseudocontact shifts” as given. However, the findings and their interpretation make a lot of sense and may be potentially also transferred to other events of ultratight protein ligand binding. However, the specific example is limited to a highly symmetric, repetitive protein and similar peptide guests, which allow a substantial conformational freedom without giving up the most powerful noncovalent interactions that would result in immediate unbinding.

A drawback of this investigation is that only NMR spectroscopic details were used to underpin a far-reaching conclusion; although the authors argue, that ITC does not yield direct information on conformational entropy, because the determination of solvens entropy is difficult, I recommend to use this technique in order to find out the overall entropy component of the entire complex formation. Solvent entropies may be estimated from calculations, and thus help to decide, if the conformational loss is effectively counterbalanced due to the internal flexibility of the system.

A second independent experimental method is highly desirable in order to support the interesting and important hypothesis put forward by the authors. If this can be done, I recommend publication in Nature Communications.

We have discussed this comment at length with colleagues, in particular with a ITC specialist in the Department of Biochemistry, UZH (Dr. I. Jeresalov). After these discussions we are convinced that it is impossible to derive meaningful data on entropy from ITC measurements, in particular for two reasons:

1) The binding affinity is too high. Femtomolar dissociation constants are outside the regime where ITC is applicable. Please note that we had to use stopped-flow techniques to derive kinetic data that were used to compute dissociation constants. Already picomolar ligands cannot be measured reliably by ITC since the associated titration curves essentially correspond to intersecting lines. One might consider doing a competition experiment but that would mean introducing more unknowns with more errors.. Please also note that reviewer 3 explicitly states that this cannot be done by ITC.

2) We do not believe that solvent entropy can be calculated with sufficient precision to allow estimating conformational entropy at reasonable precision. The solvent effects can be huge in the end.

We are not aware of any experimental technique that could describe such conformational dynamics in the ligand-bound state. In principle, this could be done by MD simulation, which, however, is not an experimental technique. But since the conformational transitions are on the ms regime it would be computationally extremely expensive (or impossible with our computing resources).

Reviewer 3:

This paper is an elegant and comprehensive study of the underlying mechanisms of the high affinity for designed Arm repeats with unstructured short peptide ligands - a fascinating and remarkable system. It represents a substantial piece of work on a simple yet complex molecular interaction, with expressed protein ligation to make samples suitable for NMR, extensive dynamic analysis, and additional biophysical and in silico data. *It was good to see the authors point out why one should not use ITC to draw conclusions about entropic effects, due to the unknown contributions from solvent*, and therefore why NMR is such an important technique.

I recommend publication with two requests-

1. They write in the abstract: "The results of this work indicate novel design principles for ultra-tight binders" Please can the authors elaborate/comment on this how one might apply their finding to protein design.
and

2. The authors write: "Transient disruptions of contacts, while the ligand is remaining tightly bound, could potentially reduce the entropic loss and therefore result in higher affinities, which might seem counter-intuitive. This concept may equally apply to any larger peptide that is tightly binding to a protein, especially with backbone and side-chains bound by multiple defined interactions as in the Armadillo system described here. It is thus not expected to be limited to interactions with repeat proteins and peptides with repetitive sequences;

however, it may require electrostatic components to ensure efficient rebinding. Importantly, it relies on a larger number of interactions such that the loss of a few interactions does not result in immediate unbinding." Can the authors speculate on how much they think this might be applicable to non-repeat proteins - they say it would need a large interface, but would that interface also need to have a high degree of intrinsic disorder? Also do they expect the effects observed to continue with increasing numbers of repeats, or is there a limit?

We have added additional remarks about the design aspects and the behavior of longer repeat proteins to the discussion (pg. 17-18). We like to remark here that some of these arguments are speculative as we have not done any comparative design studies using this principle yet.

REVIEWERS' COMMENTS

Reviewer #1 (Remarks to the Author):

The response to the query on protein expression yields is 20-60 mg/L, but 200 mg/L in the manuscript (line 513). Which value is fair?

Table S4 is still missing the units in which the tensors are reported.

The accompanying provides a lengthy response to the query of why they used the PCSs of ^{15}N spins if they are contaminated by RACS, referring to a 2021 publication. The authors argue that “adding ^{15}N PCS does not provide independent data since the ^{15}N nuclei are mostly close to the amide nitrogen when compared to the distance to the metal center” and “In fact, we have eliminated all PCS data for which the connecting line possesses a slope greater than 1.2 or smaller than 0.8, because we suspected that those couples may actually present mis-assignments of the paramagnetic species. Thereby, we also might have eliminated PCS data that contain large artefacts from ^{15}N CSA.” In the end, they state: “we prefer to stick to the original protocol.” ^{15}N PCS data may not provide independent data, but they do change the outcome of the calculations as shown in the figure provided on page 2 of the response letter. Eliminating PCS values associated with extreme slopes of the lines connecting diamagnetic and paramagnetic peaks does not address the misinterpretation of data that happen to present with the expected slope. Arguably, sticking to a flawed protocol simply is a poor decision.

Understandably, the authors are reluctant to redo calculations and figures, if calculations performed without ^{15}N PCS data would result in relatively minor corrections. It would seem appropriate, however, to point out the RACS issue in the manuscript rather than burying it in the response letter.

Reviewer #2 (Remarks to the Author):

Although I regret that the authors do not provide a second method for the characterization of the unexpected high molecular dynamics within the peptide protein complex, I accept their reasoning against ITC with too high affinity of the binding event and too low precision in determining the conformational entropy with competing solvent influence.

In light of the very positive comments of the other referees I now recommend the paper to be published after addressing their requests.

Reviewer #3 (Remarks to the Author):

I am satisfied with the changes made to the manuscript in response to the reviewers' comments and hence recommend publication.

Responses to Reviewer 1:

The response to the query on protein expression yields is 20-60 mg/L, but 200 mg/L in the manuscript (line 513). Which value is fair?

Both values are correct, one responds to the uniformly labeled protein, the lower to the segmentally labeled protein (because more purification steps were required). We have phrased this now more clearly in the Methods section on pg. 16:

While expression yields of full-length ArmRP proteins were typically very high (up to 200 mg/L), yields of the intein fusions and the (shorter) C-terminal fragments were lower, and additionally compounded by problems related to protein stability. In particular, the sensitivity of proteins to degradation due to instability of the N cap led to the development of proteins with an optimized N cap (e.g. N^AM₄C). Yields for expressed protein ligations were variable (20-60 mg/L, depending on constructs) but around 30 % for constructs with optimized Cys positions.

Table S4 is still missing the units in which the tensors are reported.

We added the units to the caption of Table S4. ($10^4 \times 1 \text{ ppm} \times \text{\AA}^3 = 10^{-32} \text{ m}^3$)

The reviewer was making a remark concerning the neglect of corrections for ¹⁵N residual anisotropic chemical shifts (RACS) in the structure calculation. As requested we clarified this matter in the manuscript now on pg. 20:

Note that no corrections for residual anisotropic chemical shifts (RACS) were done, similarly as in the originally published protocol¹⁹. However, we eliminated ¹⁵N PCS derived from peaks where the line connecting diamagnetic and paramagnetic species has a slope outside the range of 0.8 - 1.2 to remove exceptionally large RACS that would have created a local artefact in the calculation.

(there were no more request by reviewers 2 and 3).